# Nucleolin Regulates Pulmonary Artery Smooth Muscle Cell Proliferation under Hypoxia by Modulating miRNA Expression

**DOI:** 10.3390/cells12050817

**Published:** 2023-03-06

**Authors:** Jihui Lee, Hara Kang

**Affiliations:** 1Division of Life Sciences, College of Life Sciences and Bioengineering, Incheon National University, Incheon 22012, Republic of Korea; 2Institute for New Drug Development, Incheon National University, Incheon 22012, Republic of Korea

**Keywords:** pulmonary artery smooth muscle cells, nucleolin, hypoxia, microRNA

## Abstract

Hypoxia induces the abnormal proliferation of vascular smooth muscle cells (VSMCs), resulting in the pathogenesis of various vascular diseases. RNA-binding proteins (RBPs) are involved in a wide range of biological processes, including cell proliferation and responses to hypoxia. In this study, we observed that the RBP nucleolin (NCL) was downregulated by histone deacetylation in response to hypoxia. We evaluated its regulatory effects on miRNA expression under hypoxic conditions in pulmonary artery smooth muscle cells (PASMCs). miRNAs associated with NCL were assessed using RNA immunoprecipitation in PASMCs and small RNA sequencing. The expression of a set of miRNAs was increased by NCL but reduced by hypoxia-induced downregulation of NCL. The downregulation of miR-24-3p and miR-409-3p promoted PASMC proliferation under hypoxic conditions. These results clearly demonstrate the significance of NCL–miRNA interactions in the regulation of hypoxia-induced PASMC proliferation and provide insight into the therapeutic value of RBPs for vascular diseases.

## 1. Introduction

Hypoxia induces changes in gene expression which can trigger adaptive processes, such as cell proliferation or motility [1]. The alterations in numerous RNA-binding proteins (RBPs) and microRNAs (miRNAs) under hypoxic conditions have been investigated to better understand the mechanisms underlying adaptive cellular processes [2,3,4,5]. Hypoxia-regulated RBPs or miRNAs bind to specific target mRNAs for the selective regulation of expression in response to hypoxia [6,7].

Hypoxia induces structural changes in the medial compartment of the pulmonary arterial wall, including pulmonary artery smooth muscle cell (PASMC) proliferation, hypertrophy, matrix protein production, and recruitment of adventitial or circulating cells. These changes contribute to pulmonary vascular remodeling and hypertension [8,9]. Several miRNAs have been shown to modulate gene expression and PASMC function during the pathogenesis of vascular disorders under hypoxia [10,11]. However, the functions of RBPs in vascular cells under hypoxic conditions are not fully understood. Recent evidence suggests that RBPs play a role in vascular smooth muscle cells (VSMCs). For example, Hu Antigen R (HuR) contributes to the proliferation of human aortic smooth muscle cells in response to platelet-derived growth factor (PDGF) signaling [12]. In addition, the downregulation of heterogeneous nuclear ribonucleoprotein A2/B1 (HNRNPA2B1) protects against atherosclerosis by suppressing VSMC proliferation [13]. These findings suggest that RBPs contribute to the regulation of PASMC function under hypoxic conditions. Elucidating the molecular mechanism by which RBPs mediate PASMC functions under hypoxia is expected to provide a basis for the development of therapeutic strategies for pulmonary vascular diseases.

Nucleolin (NCL) is an RBP implicated in the response to hypoxia. Under hypoxia, NCL regulates the expression of matrix-metalloproteinase-9 (MMP-9) and collagen prolyl 4-hydroxylase-alpha(I) (C-P4H-alpha(I)), which are involved in ECM remodeling in human fibrosarcoma cells [14,15]. In addition to the hypoxic response, NCL is involved in a variety of biological processes, such as DNA transcription, ribosomal biogenesis, and the regulation of RNA stability [16,17,18,19,20,21]. Moreover, NCL regulates the expression of several miRNAs, including miR-15a/16, miR-21, miR-221, and miR-103. The abrogation of NCL expression affects the biogenesis of specific miRNAs, including miR-21, miR-221, and miR-103; however, the specific underlying mechanisms are unknown [22]. NCL is involved in the primary miRNA processing of miR-15a/16 through direct interactions with the microprocessor complex, DGCR8, and Drosha [23].

Interestingly, functional interactions between RBPs and miRNAs have been reported in various cancer cells [24]. Some miRNAs regulate RBP expression, and, conversely, some RBPs can modulate miRNA expression in cancer [25,26]. However, little is known about the functional relationships between RBPs and miRNAs in vascular diseases. We have hypothesized that NCL regulates the expression of hypoxia-responsive miRNAs in PASMCs and is associated with hypoxic vascular disorders. In this study, we observed that NCL levels in PASMCs are altered in response to hypoxia. To investigate the role of NCL in PASMCs in response to hypoxia, we examined its interactions with miRNAs and the functional relevance of NCL–miRNA interactions in the responses of PASMCs to hypoxia, such as their proliferation.

## 2. Results

### 2.1. Hypoxia Downregulates NCL Expression by Histone Deacetylation in PASMCs

To identify RBPs involved in the regulation of the PASMC phenotypes under hypoxia, we searched for RBPs with mRNA expression changes in the PASMCs in response to hypoxia in the next-generation RNA sequencing results from our previous studies [27]. Among the thirteen RBPs with established roles in the hypoxia-induced responses of various cells (*CIRBP*, *CPEB1*, *CPEB2*, *HNRNPA2B1*, *HNRNPL*, *IREB2*, *NCL*, *PTBP1*, *PTBP3*, *RBM3*, *TIA1*, *ZFP36*, and *ZFP36L1*), the *NCL* mRNA levels showed the greatest difference between the hypoxia-exposed and control PASMCs (Figure 1A) [14,15,28,29,30,31,32,33,34,35,36,37,38,39,40]. These results were confirmed using qRT-PCR analysis of the transcript levels in hypoxia-exposed PASMCs after 24 h (Figure 1B). Hypoxia significantly reduced the *NCL* mRNA levels to 49% of those in the control, which is consistent with the RNA sequencing data. None of the other 12 genes investigated showed significant changes in PASMCs under hypoxic conditions. A reduction in NCL protein levels following hypoxia was validated by immunoblotting (Figure 1C). The level of hypoxia-inducible factor 1-alpha (HIF1α) protein was examined by immunoblotting to confirm the hypoxic conditions in the PASMCs (Figure 1D). As expected, significant induction of HIF1α upon hypoxia was observed.

To investigate whether the hypoxia-induced decrease in NCL is specific to PASMCs, a variety of cells, including pulmonary arterial endothelial cells (PAEC), HEK293, and HeLa, were exposed to hypoxia for 24 h, and the NCL mRNA and protein levels were examined using qRT-PCR and immunoblotting. Neither the mRNA nor protein levels of the NCL were changed by hypoxia (Figure 1E,F). The induction of HIF1α with hypoxia exposure was confirmed in the PAEC, HEK293, and HeLa cells (Figure 1F). These results indicate that NCL expression is downregulated by hypoxia in the PASMCs specifically, suggesting that it plays an important role in the regulation of PASMC functions in response to hypoxia. The unique responsiveness of PASMCs to hypoxia has been reported [41]. Hypoxia increases the proliferation of PASMCs, whereas it inhibits proliferation in many other cells [41]. PASMC-specific reduction of NCL expression may contribute to inducing the unique responsiveness of PASMCs to hypoxia.

Repression of *NCL* can be mediated by histone deacetylation via histone deacetylase 1 (HDAC1) and HDAC2 [42]. We examined whether histone deacetylation is involved in the hypoxia-induced repression of *NCL* in PASMCs. PASMCs were treated with a histone deacetylase (HDAC) inhibitor, sodium butyrate (NaBu), and exposure to hypoxia. As determined by qRT-PCR (Figure 1G), the downregulation of *NCL* by hypoxia was abolished upon treatment with NaBu, suggesting that histone deacetylation is responsible for the repression of *NCL* under hypoxic conditions. To examine the role of HDAC1 or HDAC2 in the repression of NCL under hypoxia, endogenous HDAC1 and HDAC2 were reduced in PASMCs using small interfering RNAs (si-HDAC1 and si-HDAC2). The repression of NCL in response to hypoxia exposure was prevented in the PASMCs transfected with si-HDAC1 or si-HDAC2, suggesting that HDAC1 and HDAC2 are involved in the repression of *NCL* (Figure 1H). The knockdown of *HDAC1* and *HDAC2* was confirmed by qRT-PCR and immunoblotting (Figure 1I,J). According to previous studies, the HDAC1 expression levels were elevated in the lungs of patients with idiopathic pulmonary arterial hypertension and rats exposed to hypoxia, and HDAC inhibitors prevented hypoxia-induced pulmonary hypertension [43,44,45]. Therefore, hypoxia is likely to downregulate NCL expression specifically in PASMCs by histone deacetylation.

### 2.2. NCL Inhibits PASMC Proliferation

As hypoxia stimulates the proliferation of PASMCs, the role of NCL in this process was investigated. First, NCL expression in PASMCs was downregulated using siRNAs. PASMCs transfected with siRNA against NCL (si-NCL) for 24 h were stained with a Ki-67 antibody to quantify the proliferating cells (Figure 2A). Hoechst dye was then used for nuclear staining. The percentage of Ki-67-positive cells among the si-NCL-transfected cells was approximately 1.85-fold higher than that in negative control siRNA-transfected cells, suggesting that the downregulation of NCL is sufficient to promote the proliferation of PASMCs. We then overexpressed exogenous *NCL* mRNAs in PASMCs using Nucleofector (Lonza) for 48 h and examined the changes in the number of Ki-67-positive proliferating cells by immunofluorescence staining (Figure 2B). The empty pEGFP-N1 vector (Addgene) was used as a control. The percentage of proliferating cells decreased significantly to 48% when exogenous *NCL* mRNAs were overexpressed, indicating that NCL inhibited the proliferation of PASMCs. These results suggest that NCL is involved in the regulation of PASMC proliferation.

To investigate the significance of the hypoxia-induced downregulation of NCL on PASMC proliferation, we examined whether the hypoxia-induced increase in proliferation was affected by NCL overexpression (Figure 2C). PASMCs transfected with exogenous *NCL* mRNA for 48 h were exposed to hypoxia for 24 h and then stained with a Ki-67 antibody. Approximately 5.1% of the control cells were Ki-67-positive under normoxia. The percentage of proliferating cells increased to 12% under hypoxia, and this increase in cell proliferation was not detected in NCL-overexpressing cells. The results suggest that the hypoxia-induced downregulation of NCL is essential for the promotion of PASMC proliferation under hypoxic conditions.

Cell proliferation was also examined by the cell counting assay. The number of viable cells increased 24 h after the transfection of PASMCs with si-NCL (Figure 2D), whereas the number of cells decreased 24 h after the transfection of PASMCs with an NCL-overexpressing vector (Figure 2E). The number of cells increased by hypoxia decreased when NCL was overexpressed (Figure 2F). These results corroborate the observation of Ki-67 immunofluorescence staining. The efficiency of the NCL knockdown or overexpression was confirmed by qRT-PCR and immunoblotting analyses (Figure 2G,H).

### 2.3. NCL Binds to a Subset of miRNAs

In PASMCs, miRNAs play important roles in the cellular responses to hypoxia [10,11]. Interestingly, recent reports have suggested that NCL is involved in the biogenesis of several miRNAs [22,23]. Therefore, we hypothesized that hypoxia-induced changes in NCL expression result in the modulation of specific miRNAs, thereby promoting the proliferation of PASMCs. As NCL has RNA-binding properties, we searched for miRNAs associated with NCL in the PASMCs. RNAs were immunoprecipitated with an antibody against NCL or rabbit IgG as a negative control, followed by NGS-based small RNA sequencing (GSE184972). The small RNA sequencing library was made from the total RNAs from 1.5 × 10^6^ PASMCs and two samples per condition were sequenced. We identified 39 miRNAs that were specifically pulled down by NCL antibodies (≥2-fold change in pull-down samples using NCL antibodies in comparison to those using IgG, *p* < 0.05), supporting the potential role of miRNAs in NCL-induced changes in PASMCs (Table 1).

The binding of NCL to these miRNAs was validated by qRT-PCR after immunoprecipitation with an NCL antibody or rabbit IgG. The four miRNAs most highly enriched in NCL pull-down samples from the small RNA sequencing data (i.e., miR-423-3p, miR-744-5p, miR-24-3p, and miR-409-3p) showed approximately 5.5- to 33.5-fold higher levels in the pull-down samples using NCL antibodies than in those using IgG (Figure 3A). The level of miR-497-5p, which does not bind to NCL based on the results of sequencing data after immunoprecipitation, was also confirmed in the NCL pull-down sample. As expected, miR-497-5p was not enriched in either the NCL or IgG pull-down samples.

To determine whether a conserved motif exists within the sequences of the 39 potential target miRNAs of NCL, we analyzed the miRNA sequences using the motif-based sequence analysis tool MEME Suite 5.1.1 (Figure 3B). The most frequently observed motif was 5′-G/UGCUC-3′, and its position along the miRNA sequences was not identical. Since NCL is known to regulate mRNA stability by binding to a GC-rich element, it is likely that NCL binds to miRNAs with high GC contents [46].

To further confirm the interactions between the NCL and miRNAs, PASMCs were transfected with biotinylated miRNAs (bio-miR), such as bio-miR-24-3p or bio-miR-409-3p with known roles in the regulation of PASMC function, followed by affinity purification using streptavidin beads and immunoblotting with an NCL antibody (Figure 3C) [47,48,49,50]. Biotinylated *Caenorhabditis elegans* miR-67 (bio-cel-miR-67) was used as a negative control. An immunoblot analysis indicated that NCL binds to bio-miR-24-3p and bio-miR-409-3p but not to bio-cel-miR-67 (Figure 3C). The expression of exogenous biotinylated miRNAs in PASMCs transfected with bio-miR-24-3p or bio-miR-409-3p was confirmed by qRT-PCR (Figure 3D). Taken together, these results further support the hypothesis that NCL binds to specific miRNAs.

We subsequently examined whether the 5′-G/UGCUC-3′ motif is critical for the binding of NCL to miRNAs. Mutations were introduced in the motifs of bio-miR-24-3p and bio-miR-409-3p (bio-miR-24-3p mutant and bio-miR-409-3p mutant) (Figure 3B). PASMCs were transfected with these mutants or bio-cel-miR-67, followed by a pull-down assay (Figure 3C). Mutations in the motif abrogate the binding of NCL to miRNAs, suggesting that the 5′-G/UGCUC-3′ motif serves as an NCL-binding site. These results suggest that NCL binds to specific miRNAs via this binding site and selectively regulates miRNA expression.

### 2.4. NCL Modulates miRNA Expression

We examined whether NCL affects the expression levels of miRNAs that bind to NCL. PASMCs were transfected with negative control siRNA (control) or si-NCL for 24 h and miRNA levels were measured by qRT-PCR. When NCL was downregulated by siRNAs, the expression levels of the miRNAs, including miR-423-3p, miR-744-5p, miR-24-3p, and miR-409-3p, were reduced by 51–70% compared with levels in the control (Figure 4A). We subsequently overexpressed exogenous *NCL* mRNAs in PASMCs using Nucleofector (Lonza) for 24 h and examined the levels of these miRNAs (Figure 4B). The expression levels of four miRNAs were 1.3–2.8-fold higher in the *NCL*-overexpressed cells than in the control cells transfected with the empty pEGFP-N1 vector (control). The level of miR-497-5p, which does not bind to NCL, was not affected by the knockdown or overexpression of NCL. These results suggest that NCL binds to certain miRNAs and regulates their expression.

Previous studies have shown that NCL can affect the biogenesis of miRNAs or promote targeted mRNA degradation of miRNAs via interactions with miRNA-associated proteins, such as DGCR8, Drosha, or Ago2 [23,51,52]. Thus, we investigated their interactions in PASMCs. The cellular NCL from the PASMCs was immunoprecipitated with an NCL antibody or IgG control and analyzed by Western blotting with antibodies against DGCR8, Ago2, or NCL (Figure 4C,D). Conversely, lysates of PASMCs were immunoprecipitated with antibodies against DGCR8, Ago2, or IgG, and Western blotting was used to determine whether NCL was present in the pull-down (Figure 4E,F). We found that NCL binds to DGCR8 and Ago2 in PASMCs. These results further support the role of NCL in the regulation of miRNA expression and activity.

### 2.5. NCL Mediates Hypoxia-Induced Regulation of miRNA Expression

Given that hypoxia downregulates NCL expression, miRNAs regulated by NCL are expected to show lower expression under hypoxic conditions. We examined the changes in the expression levels of miRNAs, including miR-423-3p, miR-744-5p, miR-24-3p, and miR-409-3p, under hypoxia using qRT-PCR. The expression levels of miR-423-3p, miR-744-5p, miR-24-3p, and miR-409-3p were all reduced by exposure to hypoxia for 24 h (Figure 5A). These results suggest that the hypoxia-induced downregulation of NCL is responsible for the reduced expression of certain miRNAs.

To determine whether the modulation of NCL expression influences the hypoxia-induced regulation of miRNA expression, we overexpressed exogenous *NCL* mRNAs in PASMCs using Nucleofector for 24 h before hypoxia exposure and examined the levels of miR-423-3p, miR-744-5p, miR-24-3p, and miR-409-3p using qRT-PCR. When NCL was overexpressed, the hypoxia-induced reduction in the miRNA levels was restored (Figure 5B). These results indicate that the hypoxia-induced modulation of NCL expression controls the expression of certain miRNAs.

### 2.6. Downregulation of miR-24-3p and miR-409-3p Promotes PASMC Proliferation under Hypoxia

We determined the biological consequences of NCL-mediated miRNA regulation in hypoxic PASMCs. As NCL was observed to regulate PASMC proliferation (Figure 2), we further examined whether miRNAs regulated by NCL, such as miR-24-3p and miR-409-3p, would affect PASMC proliferation. PASMCs were transfected with control, miR-24-3p mimic, miR-409-3p mimic, miR-24-3p antisense inhibitor RNA (anti-miR-24-3p), or anti-miR-409-3p and stained with a Ki-67 antibody. miR-24-3p and miR-409-3p mimics significantly reduced the number of Ki-67-positive proliferating cells by 61% and 70%, respectively, compared to cell counts in the control (Figure 6A). Conversely, cells transfected with anti-miR-24-3p or anti-miR-409-3p showed increased numbers of proliferating cells (i.e., 1.83-fold or 1.62-fold higher than counts in the control) (Figure 6B). These results demonstrate that the downregulation of miR-24-3p and miR-409-3p is required to promote PASMC proliferation.

To examine whether the modulation of miR-24-3p or miR-409-3p affects the hypoxia-induced proliferative response of PASMCs, PASMCs were transfected with control, miR-24-3p mimic, or miR-409-3p mimic prior to exposure to hypoxia and stained with a Ki-67 antibody. The hypoxia-induced increase in PASMC proliferation was inhibited in cells transfected with miR-24-3p mimic or miR-409-3p mimic (Figure 6C). Therefore, it is likely that the hypoxia-induced downregulation of NCL promotes PASMC proliferation by downregulating a subset of miRNAs, including miR-24-3p and miR-409-3p.

We also carried out a cell counting assay to determine cell proliferation. Consistent with the results of Ki-67 immunostaining, the rate of proliferation decreased in PASMCs transfected with miR-24-3p, or miR-409-3p mimics (Figure 6D). In contrast, the proliferation of PASMCs was promoted by anti-miR-24-3p, or anti-miR-409-3p (Figure 6E). The hypoxia-induced increase in cell proliferation was inhibited by miR-24-3p or miR-409-3p (Figure 6F). Therefore, the downregulation of miR-24-3p and miR-409-3p is essential for promoting PASMC proliferation under hypoxia. To confirm the overexpression or downregulation of miR-24-3p and miR-409-3p, their levels were measured in the PASMCs at 24 h after transfection with the control, miR-24-3p mimic, miR-409-3p mimic, anti-miR-24-3p, or anti-miR-409-3p (Figure 6G,H).

## 3. Discussion

RBPs are important regulators of gene expression via post-transcriptional regulation. Under hypoxia, RBPs regulate the expression of hypoxia-inducible genes. However, the role of RBPs in the functions of PASMCs under hypoxic conditions and the molecular mechanisms underlying their effects are not yet fully understood. In this study, we identified NCL as an essential regulator of PASMC proliferation under hypoxia and characterized its molecular function. miRNAs act as critical mediators of the response to hypoxia in PASMCs. As recent studies have revealed that NCL is involved in the biogenesis of specific miRNAs, we evaluated NCL–miRNA interactions in PASMCs by immunoprecipitation and small RNA sequencing. Thirty-nine miRNAs enriched in NCL pull-down were identified. We further found that the hypoxia-induced downregulation of NCL affects the expression of these miRNAs and demonstrated that NCL-mediated miRNA regulation induces the proliferation of PASMCs under hypoxic conditions.

Given that an RBP deficiency is associated with cardiovascular developmental defects, RBPs may play a critical role in maintaining cardiovascular health. Recently, RBPs have been implicated in systematic cardiovascular disease via the post-transcriptional regulation of target genes. For example, quaking (QKI) in VSMCs binds to myocardin and derives alternative splicing in response to vessel injury [53]. The identification of the modulation of NCL in PASMCs under hypoxia extends our understanding of the functions of RBPs in vascular conditions and provides new targets for the treatment of vascular diseases.

NCL expression has been shown to be regulated by HuR and several miRNAs. HuR interacts with the 3′UTR of *NCL* and promotes its translation, whereas miR-494, miR-194, and miR-206 suppress NCL expression [54,55]. In this study, we observed that both the mRNA and protein levels of NCL were significantly reduced by hypoxia. We examined whether the levels of miR-494, miR-194, or miR-206 increased in hypoxia-exposed PASMCs to suppress NCL expression. Our previously generated small RNA sequencing data showed that miR-494 and miR-194 levels did not change in response to hypoxia, and the expression of miR-206 was not determined [27]. It is therefore unlikely that these three miRNAs are responsible for the decrease in NCL expression in PASMCs under hypoxia. Rather, we elucidated the role of HDAC in the transcriptional repression of NCL under hypoxia. NCL has been linked to a variety of pathologies, including carcinogenesis, and thus, elucidating the regulatory mechanisms underlying its expression should provide a basis for the development of new therapeutic strategies for a variety of diseases, including hypoxia-induced vascular diseases.

There is emerging evidence of the involvement of RBPs in the regulation of miRNA biogenesis. For example, HNRNPA1 promotes Drosha cleavage by restructuring pri-miR-18a [56]. NCL has also been reported to enhance the maturation of specific miRNAs, including miR-21, miR-221, and miR-222, and is consequently involved in the pathogenesis of cancer [22]. We have demonstrated that NCL controls the fate of miRNAs in response to hypoxia in PASMCs. NCL binds to and regulates certain miRNAs, particularly those that contain the 5′-G/UGCUC-3′ sequence. We have provided the first evidence to elucidate the biochemical interactions between NCL and miRNAs in PASMCs and their role in the proliferation of PASMCs. The regulation of miRNA expression by NCL is essential for PASMC responses to hypoxic conditions.

The proliferation of VSMCs is a hallmark of several vascular pathologies as well as hypoxia-induced remodeling [41,57]. Multiple miRNAs involved in the proliferation of VSMCs have also been explored [58,59,60]. For example, miR-24 inhibits high glucose-stimulated VSMC proliferation by targeting high mobility group box-1 (HMGB1) [49]. Overexpression of miR-24 reduced neointimal hyperplasia and VSMC proliferation by inhibiting the Wnt4 signaling pathway [47]. In addition, miR-24 suppressed the platelet-derived growth factor-BB (PDGF-BB) signaling pathway by decreasing the expression levels of activator protein 1 (AP-1) and the PDGF-receptor (PDGF-R), resulting in the inhibition of VSMC proliferation and vascular remodeling [50]. The results imply that miR-24 may also regulate VSMC proliferation under hypoxia. While few previous studies have explored the function of miR-409 in VSMCs, decreased miR-409 expression levels were observed during high phosphate-induced vascular calcification, triggering VSMC de-differentiation [48]. This finding suggests that miR-409 may be involved in the regulation of VSMC proliferation. We have demonstrated that the target miRNAs of NCL influence the proliferation of PASMCs. For example, miR-24-3p and miR-409-3p inhibit PASMC proliferation and their overexpression further prevents hypoxia-induced proliferation. These results add a layer of valuable information about a specific set of miRNAs that regulate the proliferation of PASMCs. In addition, as the target miRNA level is regulated by the level of NCL expression, it is clear that NCL–miRNA interactions are essential for the regulation of PASMC proliferation. As miRNAs are potent regulators of cellular function in pathophysiological conditions, our illustration of NCL–miRNA interactions and the role of NCL in PASMC functions via the regulation of miRNAs improves our general understanding of the mechanisms underlying the pathogenesis of vascular conditions related to hypoxia. To explore the potential therapeutic benefits of NCL or interacting miRNAs on pulmonary hypertension, it is necessary to investigate whether modulation of NCL or interacting miRNAs is effective in attenuating pulmonary vascular remodeling in animal models, such as a chronic hypoxia-induced rat model.

## 4. Conclusions

In this study, we provide clear evidence for the role of the RBP nucleolin (NCL) in hypoxia-induced PASMC proliferation. NCL is downregulated by histone deacetylation under hypoxic conditions in PASMCs, which consequently promotes PASMC proliferation. Furthermore, we demonstrated that these effects of NCL are mediated by interactions with a subset of miRNAs using immunoprecipitation and NGS-based small RNA sequencing. Thirty-nine miRNAs were found to be enriched in NCL pull-down, and NCL regulates particular miRNA expressions via the 5′-G/UGCUC-3′ binding sites. Hypoxia-mediated regulation of NCL affects miRNA expression, and these miRNAs, such as miR-24-3p and miR-409-3p, are involved in the proliferation of PASMCs under hypoxia. Collectively, the identification of NCL-miRNA interactions in hypoxia-induced PASMC proliferation provides a basis for further studies of the molecular mechanisms underlying vascular diseases.

## 5. Materials and Methods

### 5.1. Cell Culture and Hypoxia

Human primary pulmonary artery smooth muscle cells (PASMCs) were purchased from Lonza (CC-2581) and were maintained in Sm-GM2 medium (Lonza, Basel, Switzerland) containing 5% fetal bovine serum (FBS). For hypoxia, the cells were placed in fresh medium and incubated in a sealed modular incubator chamber (Billups-rothenberg Inc., San Diego, CA, USA) for 24 h at 37 °C after flushing with a mixture of 5% CO_2_, 1% O_2_ and 94% N_2_ for 4 min.

### 5.2. Sodium Butyrate (NaBu) Treatment

NaBu was purchased from Sigma-Aldrich (St. Louis, MO, USA, #B5887). The cells were treated with 10 mM NaBu for 24 h.

### 5.3. Quantitative Reverse Transcriptase-PCR (qRT-PCR)

Quantitative analysis of the change in expression levels was performed using real-time PCR. The mRNA levels were normalized to 18S rRNA. The primers used were as follows: 18S rRNA, 5′-GTAACCCGTTGAACCCCATT-3′ and 5′-CCATCCAATCGGTAGTAGCG-3; CIRBP, 5′-CTTTTTGTTGGAGGGCTGAG-3′ and 5′-CTTGCCTGCCTGGTCTACTC-3′; CPEB1, 5′-TCTGCCCTTCCTGTCTCTGT-3′ and 5′-TATGCTGAAGGGGTCTTTGG-3′; CPEB2, 5′-GCGAGTTGCTTTCTCCAATC-3′ and 5′-CCTGGCATTCATCACACATC-3′; HNRNPA2B1, 5′-GGCTACGGAGGTGGTTATGA-3′ and 5′- ATAACCCCCACTTCCTCCAC -3′; HNRNPL, 5′-AGATCACCCCGCAGAATATG -3′ and 5′-CAAGCCATAGACCATGAGCA -3′; IREB2, 5′-GCACCGGATTCAGTTTTGTT-3′ and 5′-CTTAGCGGCAGCACTATTCC-3′; NCL, 5′-GAAGGAAATGGCCAAACAGA-3′ and 5′-ACGCTTTCTCCAGGTCTTCCA-3′; PTBP1, 5′-ACGGACCGTTTATCATGAGC-3′ and 5′-GTTTTTCCCCTTCAGCATCA-3′; PTBP3, 5′-CATTCCTGGGGCTAGTGGTA-3′ and 5′-CCATCTGAACCAAGGCATTT-3′; RBM3, 5′-CAGGCACTGGAAGACCACTT-3′ and 5′-CTCTCATGGCAACTGAAGCA-3′; TIA1, 5′-TGCTATTGGGGCAAAGAAAC-3′ and 5′-GCGGTTGCACTCCATAATTT-3′; ZFP36, 5′-TCCACAACCCTAGCGAAGAC-3′ and 5′-GAGAAGGCAGAGGGTGACAG-3′; and ZFP36L1, 5′-GAGGAAAACGGTGCCTGTAA-3′ and 5′-CTCTTCAGCGTTGTGGATGA-3′. For the quantification of mature miRNAs, such as miR-423-3p (MS00004179), miR-744-5p (MS00010549), miR-24-3p (MS00006552), miR-409-3p (MS00006895), and miR-497-5p (MS00004361), the miScript PCR assay kit (#218073, Qiagen, Hilden, Germany) was used according to the manufacturer’s instructions. Data analysis was performed using a comparative C_T_ method in the Bio-Rad software 3.1. The levels of the miRNAs were normalized to U6 small nuclear RNA or U61 small nucleolar RNA (SNORD61). Three experiments were performed in triplicate, and the mean results with standard errors are presented.

### 5.4. miRNA Mimics and Anti-miRNA Oligonucleotides

Chemically modified double-stranded RNAs designed to mimic the endogenous mature miR-24-3p and miR-409-3p were purchased from Genolution Pharmaceuticals (Seoul, Republic of Korea). Antisense inhibitor RNAs (anti-miR-24-3p and anti-miR-409-3p) and negative control miRNA were purchased from Bioneer (Daejeon, Republic of Korea). The miRNA mimics and anti-miRNA oligonucleotides were transfected at 5 nM and 25 nM, respectively, using RNAi Max (Invitrogen, Carlsbad, CA, USA) according to the manufacturer’s protocol.

### 5.5. RNA Interference

Small interfering RNA (siRNA) duplexes were synthesized by Genolution Pharmaceuticals (Seoul, Republic of Korea) and Integrated DNA Technologies (Coralville, IA, USA). NCL siRNA (si-NCL): 5′-GGAUAGUUACUGACCGGGA-3′, HDAC1 siRNA (si-HDAC1): 5′-AGUUUCCUUUUUGAGAUACUAUUTT-3′, and HDAC2 siRNA (si-HDAC2): 5′-GAAUUUCUAUUCGAGCAUCAGACAA-3′. Negative control siRNA (Genolution) was used as a control. The cells were transfected with 100 nM si-NCL, 25 nM si-HDAC1, or si-HDAC2 using RNAi Max (Invitrogen, Carlsbad, CA, USA) according to the manufacturer’s protocol.

### 5.6. Immunoprecipitation

PASMC lysates were prepared in lysis buffer (20 mM Tris–HCl [pH 7.5], 100 mM KCl, 5 mM MgCl_2_, and 0.5% NP-40) containing a protease inhibitor cocktail (#11697498001, Roche, Basel, Switzerland). The lysates were precleared with Dynabeads^TM^ Protein G (#10007D, Invitrogen, Carlsbad, CA, USA) at 4 °C with gentle rotation for 1 h. The precleared lysates were incubated with Dynabeads^TM^ Protein G coated with 2 μg each of rabbit anti-nucleolin antibody (#ab22758, Abcam, Cambridge, UK) or rabbit IgG (#2729, Cell Signaling Technology, Danvers, MA, USA) at 4°C for overnight. For DGCR8 IP and Ago2 IP, 5 μg of mouse anti-DGCR8 antibody (#60084-1-Ig, Proteintech, Rosemont, IL, USA) and mouse anti-Ago2 antibody (#ab57113, Abcam) were used, respectively. A reaction containing mouse IgG (#sc-2025, Santa Cruz Biotechnology, Dallas, TX, USA) served as a negative control. Unbound materials were washed off using NT2 buffer (50 mM Tris–HCl [pH 7.5], 150 mM NaCl, 1 mM MgCl_2_, and 0.05% NP-40). All collected protein complexes were eluted with 2X Laemmli sample buffer supplemented with β-mercaptoethanol and boiled. The boiled supernatants and input (2%) samples were resolved by SDS-PAGE and analyzed by immunoblotting with the anti-NCL antibody (#ab22758), anti-DGCR8 antibody (#60084-1-Ig), or anti-Ago2 antibody (#ab57113).

### 5.7. RNA Immunoprecipitation

PASMC lysates were prepared in lysis buffer (20 mM Tris–HCl (pH 7.5), 100 mM KCl, 5 mM MgCl_2_, and 0.5% NP-40) containing 40 U/μL RiboLock RNase Inhibitor (#EO0381, Thermo Fisher Scientific, Waltham, MA, USA) and a protease inhibitor cocktail (#11697498001, Roche, Basel, Switzerland). The lysates were incubated with Dynabeads^TM^ Protein G (#10007D, Invitrogen, Carlsbad, CA, USA) coated with 2 μg each of rabbit anti-NCL antibody (#ab22758, Abcam, Cambridge, UK) or rabbit IgG (#2729, Cell Signaling Technology, Danvers, MA, USA) at 4 °C for 2 h. Unbound materials were washed off using NT2 buffer (50 mM Tris–HCl (pH 7.5), 150mM NaCl, 1 mM MgCl_2,_ and 0.05% NP-40). The pellet was subsequently incubated with NT2 buffer containing RNase-free Dnase I (1 U/μL) (#EN0521, Thermo Fisher Scientific, Waltham, MA, USA) at 30 °C for 15 min and NT2 buffer containing 0.1% SDS and 0.1 mg/mL Proteinase K (#25530049, Thermo Fisher Scientific, Waltham, MA, USA) at 55 °C for 15 min. The RNA was extracted using Trizol in the presence of glycoblue (#AM9515, Thermo Fisher Scientific, Waltham, MA, USA) and analyzed by NGS-based small RNA sequencing or qRT-PCR.

### 5.8. NGS-Based Small RNA Sequencing

The extracted RNA was qualified using an Agilent 2100 Bioanalyzer (Agilent Technologies, Santa Clara, CA, USA). cDNA libraries were constructed with the NEBNext Multiplex small RNA library prep kit (NEB, Ipswich, MA, USA) using the total RNA from RNA immunoprecipitation, according to the manufacturer’s instructions. Briefly, adapter ligation, reverse transcription, PCR amplification, and purification using a QIAquick PCR Purification Kit (Qiagen) and AMPure XP beads (Beckman Coulter, Brea, CA, USA) were conducted to generate a library product. The yield and size distribution of the small RNA libraries were assessed by high-sensitivity DNA analysis on an Agilent 2100 Bioanalyzer. High-throughput sequences were produced by single-end 75 sequencing using the NextSeq 500 system (Illumina, San Diego, CA, USA). Sequence reads were mapped using the Bowtie 2 software tool to obtain the BAM file (alignment file). A mature miRNA sequence was used as a reference for mapping. Read counts mapped onto the mature miRNA sequence were extracted from the alignment file using bedtools (v2.25.0) and Bioconductor, which uses R (version 3.2.2) statistical programming language (R development Core Team, 2011). The read counts were then used to determine the expression levels of miRNAs. The quantile normalization method was used to compare samples.

### 5.9. Biotinylated miRNA Pull-Down Assay

The 3′-biotinylated miR-24-3p mimic (bio-miR-24-3p), 3′-biotinylated miR-409-3p mimic (bio-miR-409-3p), 3′-biotinylated miR-24-3p mutant (bio-miR-24-3p mutant), 3′-biotinylated miR-409-3p mutant (bio-miR-409-3p mutant), and 3′-biotinylated control *Caenorhabditis elegans* miR-67 mimic (bio-cel-miR-67) were synthesized by Integrated DNA Technologies (Coralville, IA, USA). PASMCs were transfected with 150 nM bio-miR-24-3p, bio-miR-409-3p, bio-miR-24-3p mutant, bio-miR-409-3p mutant, or bio-cel-miR-67 mimic using RNAi MAX (Invitrogen, Carlsbad, CA, USA). Twenty-four hours later, the cells were trypsinized and lysed in lysis buffer (20 mM Tris–HCl (pH 7.5), 100 mM KCl, 5 mM MgCl_2,_ and 0.5% NP-40) containing 40 U/μL RiboLock RNase Inhibitor (#EO0381, Thermo Fisher Scientific, Waltham, MA, USA) and a protease inhibitor cocktail (#11697498001, Roche, Basel, Switzerland) on ice for 20 min and centrifuged at 12,000 rpm for 10 min at 4 °C. The lysates were incubated with Streptavidin Mag Sepharose (#GE28-9857-38, Sigma-Aldrich, St. Louis, MO, USA) at 4 °C for 4 h, and unbound materials were washed off using NT2 buffer. The pull-down sample was boiled in 2X Laemmli sample buffer supplemented with β-mercaptoethanol. The boiled pull-down and input (1%) samples were resolved by SDS-PAGE and analyzed by immunoblotting with the anti-NCL antibody (#ab22758, Abcam).

### 5.10. NCL Expression Plasmid

The plasmid GFP-NCL was a gift from Michael Kastan (Addgene plasmid #28176; http://n2t.net/addgene:28176; RRID: Addgene_28176) [61]. PASMCs were transfected with 1 μg of the GFP-NCL or the empty pEGFP-N1 vector (Addgene, Watertown, MA, USA) using the P1 Primary Cell 4D-Nucleofector^TM^ X kit (Lonza, Basel, Switzerland) according to the manufacturer’s protocol.

### 5.11. Immunoblotting

Cells were lysed in TNE buffer (50 mM Tris–HCl (pH 7.4), 100 mM NaCl, and 0.1 mM EDTA) and total cell lysates were separated by SDS-PAGE, transferred to PVDF membranes, immunoblotted with antibodies, and visualized using an enhanced chemiluminescence detection system (Bio-Rad Laboratories, Hercules, CA, USA). Antibodies against NCL (#ab22758), HIF1α (#ab2185), and Ago2 (#ab57113) were purchased from Abcam (Cambridge, UK). An anti-β-actin antibody (#sc47778), anti-HDAC1 antibody (#sc81598), anti-HDAC2 antibody (#sc-81599), and anti-DGCR8 antibody (#60084-1-Ig) were purchased from Santa Cruz Biotechnology (Dallas, TX, USA) and Proteintech (Rosemont, IL, USA).

### 5.12. Immunofluorescence Staining

Equal amounts of PASMCs were seeded in chamber well slides and transfected with control mimic, miR-24-3p, miR-409-3p, anti-miR-24-3p, or anti-miR-409-3p. Cells were exposed to normoxia or hypoxia and then fixed in 2% paraformaldehyde, blocked in 3% BSA in PBS, and permeabilized in 0.1% Triton X-100 (Sigma-Aldrich, St. Louis, MO, USA) in PBS. The slides were sequentially probed with rabbit anti-human Ki-67 antibody (#ab16667, Abcam) and goat anti-rabbit IgG (H + L) cross-adsorbed secondary antibody Alexa flour 488 (#A-11008, Thermo Fisher Scientific). Nuclei were stained with Hoechst 33342 (#62249, Thermo Fisher Scientific). The slides were imaged by a Zeiss Axio Imager Z1 microscope (Oberkochen, Germany). At least 2000 cells were counted per condition, and the percentages of Ki-67-positive cells were presented. The results are the mean ± S.E. for triplicate assays.

### 5.13. Cell Counting Assay

Equal amounts of PASMCs were seeded in plates and transfected with negative control siRNA, si-NCL, pEGFP-N1 vector, GFP-NCL, control mimic, miR-24-3p mimic, miR-409-3p mimic, anti-miR-24-3p, or anti-miR-409-3p. The cells were trypsinized and manually counted using a hemocytometer. The total cell numbers were compared and presented as a fold change.

### 5.14. Statistical Analysis

All experiments were performed with at least three independent repetitions. The results were presented as the mean with standard error. Statistical analyses were performed by an analysis of variance followed by Student’s *t*-test, multiple *t*-test, one-way ANOVA, or two-way ANOVA using Prism 8 software (GraphPad Software Inc., San Diego, CA, USA). *p*-values of <0.05 were considered significant and are indicated with asterisks. *, **, ***, and **** represent *p*-values less than 0.05, 0.005, 0.0005, and 0.0001, respectively.

### 5.15. Accession Numbers

The RNA sequencing dataset generated during the current study is available from the corresponding author on reasonable request. The accession numbers for the data reported in this paper are GEO: GSE184972.

## Figures and Tables

**Figure 1 cells-12-00817-f001:**
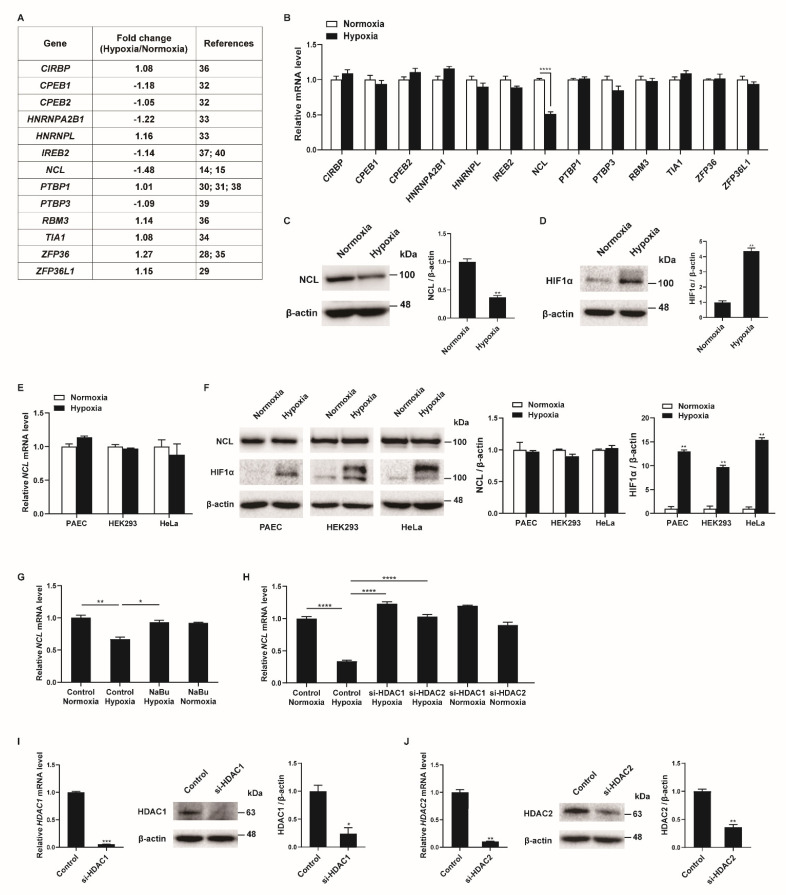
Hypoxia downregulates NCL in PASMCs. (**A**) Expression levels of RBPs were compared in PASMCs exposed to normoxia or hypoxia by NGS-based RNA sequencing from our previous study [27]. (**B**) Levels of the indicated mRNAs relative to 18S rRNA measured by qRT−PCR in PASMCs exposed to normoxia or hypoxia for 24 h. Data represent the mean ± S.E. of triplicate measurements of three independent experiments. (**C**,**D**) Immunoblot analysis of lysates from PASMCs exposed to normoxia or hypoxia for 24 h with anti−NCL, anti−HIF1α, or anti-β−actin antibody. By densitometry, relative amounts of NCL (**C**) and HIF1α (**D**) normalized to β−actin were quantitated. (**E**) Levels of NCL mRNAs relative to 18S rRNA measured by qRT−PCR in PAEC, HEK293, and HeLa exposed to normoxia or hypoxia for 24 h. Data represent the mean ± S.E. of triplicate measurements of three independent experiments. (**F**) Immunoblot analysis of lysates from PAEC, HEK293, and HeLa exposed to normoxia or hypoxia for 24 h using anti−NCL, anti−HIF1α, or anti−β−actin antibodies. By densitometry, relative amounts of NCL or HIF1α normalized to β−actin were quantitated. (**G**) Levels of *NCL* mRNAs relative to 18S rRNA measured by qRT−PCR in PASMCs treated with HDAC inhibitor NaBu under normoxia or hypoxia. (**H**) PASMCs were transfected with control or siRNAs against HDAC1 or HDAC2 for 24 h, followed by hypoxia exposure for 24 h. Levels of *NCL* mRNAs relative to 18S rRNA were quantitated. Data represent the mean ± S.E. of triplicate measurements of three independent experiments. (**I**,**J**) (Left panel) Levels of *HDAC1* (**I**) or *HDAC2* (**J**) mRNAs relative to 18S rRNA measured by qRT-PCR in PASMCs 24 h after transfection with control or siRNAs. The data represent the mean ± S.E. of triplicate measurements of three independent experiments. (Right panel) Immunoblot analysis of lysates from PASMCs transfected with control, si−HDAC1 (**I**), or si−HDAC2 (**J**) with antibodies against HDAC1, HDAC2, or β−actin. By densitometry, relative amounts of HDAC1 and HDAC2 normalized to β−actin were quantitated. Statistical analyses were performed using two−way ANOVA Sidak’s multiple comparisons test (**B**), two−tailed unpaired Student’s *t*-test (**C**,**D**,**I**,**J**), multiple *t*-test (**E**,**F**), and one−way ANOVA Tukey’s multiple comparisons test (**G**,**H**). *, *p* < 0.05; **, *p* < 0.005; ***, *p* < 0.0005; ****, *p* < 0.0001.

**Figure 2 cells-12-00817-f002:**
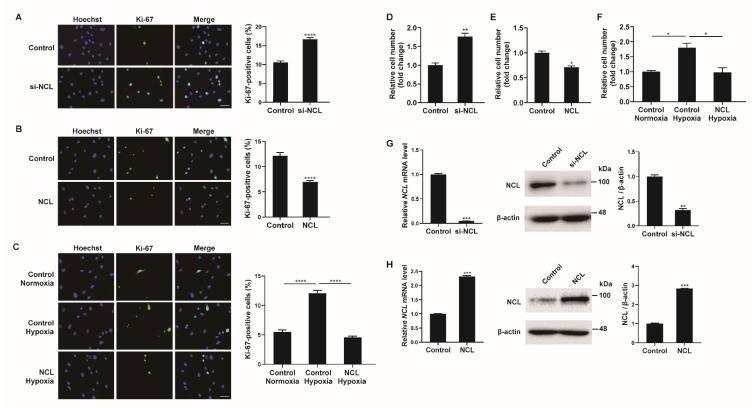
NCL regulates PASMC proliferation. (**A**) Representative images of Ki−67 immunostaining of PASMCs transfected with the control or siRNA against NCL (si−NCL) and calculation of the Ki−67 index. (**B**,**C**) Immunostaining analysis of PASMCs nucleofected with the empty vector control or NCL−expressing plasmids at a steady state (**B**) and after hypoxia exposure (**C**). Approximately 200 cells from at least 10 independent fields were counted for each condition, and Ki−67-positive cells are presented as a percentage of the total population. The results are the mean ± S.E. for three independent assays. Scale bar represents 50 μm. (**D**,**E**) PASMCs were transfected with control, si-NCL, an empty vector control, or an NCL−expressing vector. Cells were trypsinized and counted using a hemacytometer. The relative number of cells was shown as a fold change. (**F**) PASMCs were transfected with an empty vector control or an NCL-expressing vector, followed by hypoxia exposure and cell counting assay. (**G**,**H**) (Left panel) Expression levels of *NCL* mRNA normalized to 18S rRNA measured by qRT-PCR in PASMCs transfected with control or si−NCL for 24 h (**G**), or nucleofected with control or NCL−expressing plasmid for 24 h (**H**). (Right panel) Immunoblot analysis of lysates from NCL knockdown (**G**) or overexpressed (**H**) PASMCs with antibodies against NCL or β−actin. By densitometry, relative amounts of NCL protein normalized to β−actin were quantitated. Statistical analyses were performed using the two-tailed unpaired Student’s *t*-test (**A**,**B**,**D**,**E**,**G**,**H**) and one−way ANOVA Tukey’s multiple comparisons test (**C**,**F**). *, *p* < 0.05; **, *p* < 0.005; ***, *p* < 0.0005; ****, *p* < 0.0001.

**Figure 3 cells-12-00817-f003:**
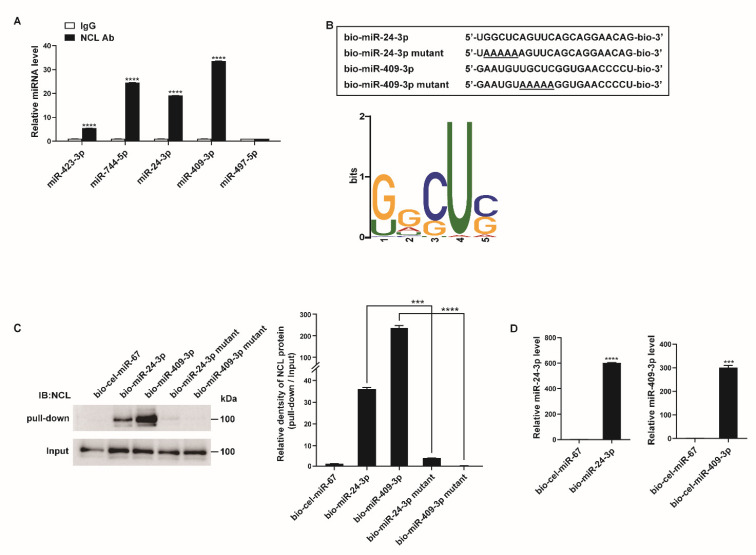
NCL binds to a subset of miRNAs. (**A**) PASMCs were subjected to RNA immunoprecipitation assay using an anti−NCL antibody. qRT−PCR was then performed to measure the enrichment of the indicated miRNAs. Levels of miRNAs relative to SNORD61 were quantitated. The data represent the mean ± S.E. of triplicate measurements of three independent experiments. (**B**) (**Upper panel**) Schematic diagram of miRNA wild type and mutant sequences. Underlined characters indicate mutations introduced. (**Lower panel**) Sequence logo representing the conserved motif present in miRNAs associated with NCL. The overall height of each stack indicates the sequence conservation at that position (measured in bits). (**C**) Immunoblot analysis using anti−NCL antibodies for the pull−down and input (1%) samples from PASMCs transfected with biotinylated miRNAs or biotinylated miRNA mutants containing the mutated motif sequence. (**D**) Levels of miR-24-3p or miR-409-3p relative to U6 snRNA measured by qRT−PCR in PASMCs transfected with bio-cel-miR-67, bio-miR-24-3p, or bio-miR-409-3p. The data represent the mean ± S.E. of triplicate measurements of three independent experiments. Statistical analyses were performed using two−way ANOVA Sidak’s multiple comparisons test (**A**), one−way ANOVA Tukey’s multiple comparisons test (**C**), and two−tailed unpaired Student’s *t*-test (**D**). ***, *p* < 0.0005; ****, *p* < 0.0001.

**Figure 4 cells-12-00817-f004:**
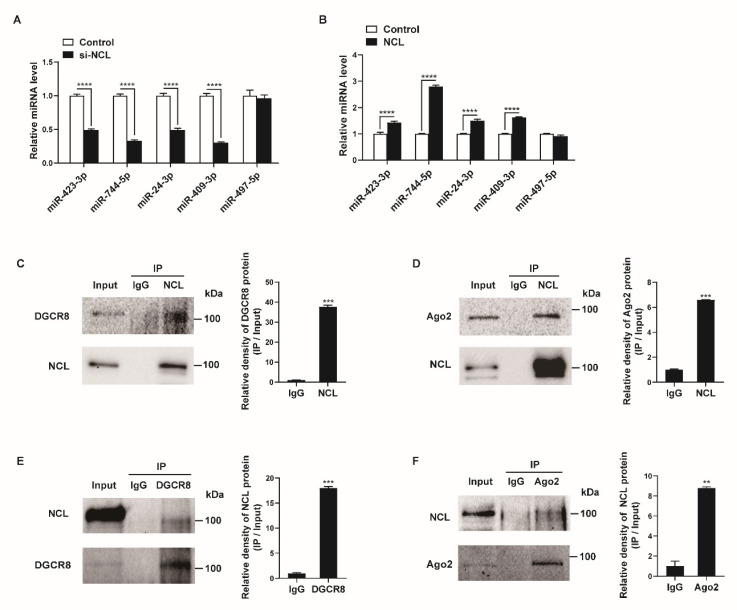
NCL expression affects specific miRNA levels. (**A**,**B**) Levels of the indicated miRNAs relative to U6 snRNA measured by qRT−PCR in PASMCs transfected with control or si−NCL for 24 h (**A**) or nucleofected with the control or NCL−expressing plasmid for 24 h (**B**). The data represent the mean ± S.E. of triplicate measurements of three independent experiments. (**C**,**D**) Immunoprecipitation assays from PASMC lysates were performed using an anti-NCL antibody or IgG. The presence of DGCR8, Ago2, and NCL in the pull-down and input (2%) samples was monitored by immunoblotting. Input: immunoblots showing the abundance of DGCR8, Ago2, or NCL in the total protein extracts. IP: IP products precipitated by the anti−NCL, anti−DGCR8, anti−Ago2 antibody, or IgG. (**E**,**F**) Immunoprecipitation assays from PASMC lysates were carried out using an anti-DGCR8 antibody, anti−Ago2 antibody, or IgG. The presence of NCL, DGCR8, and Ago2 in the immunoprecipitates and input (2%) samples was monitored by immunoblotting. Statistical analyses were performed using two−way ANOVA Sidak’s multiple comparisons test (**A**,**B**) and two−tailed unpaired Student’s *t*-test (**C**,**F**). **, *p* < 0.005; ***, *p* < 0.0005; ****, *p* < 0.0001.

**Figure 5 cells-12-00817-f005:**
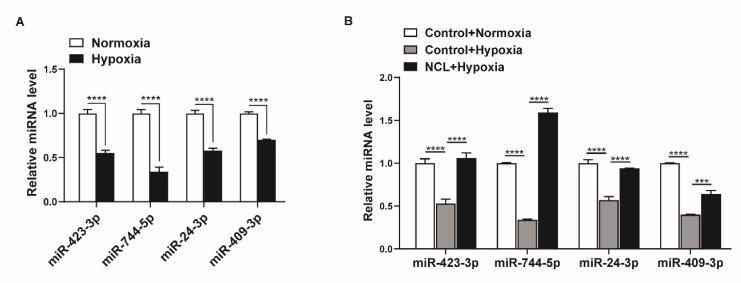
NCL is responsible for the hypoxia-induced downregulation of specific miRNAs. (**A**) Levels of miRNAs relative to U6 snRNA measured by qRT−PCR in PASMCs exposed to hypoxia for 24 h. The data represent the mean ± S.E. of triplicate measurements of three independent experiments. (**B**) Levels of miRNAs relative to U6 snRNA measured by qRT-PCR in NCL−overexpressing PASMCs after exposure to hypoxia for 24 h. The data represent the mean ± S.E. of triplicate measurements of three independent experiments. Statistical analyses were performed using two−way ANOVA Sidak’s multiple comparisons test (**A**,**B**). ***, *p* < 0.0005; ****, *p* < 0.0001.

**Figure 6 cells-12-00817-f006:**
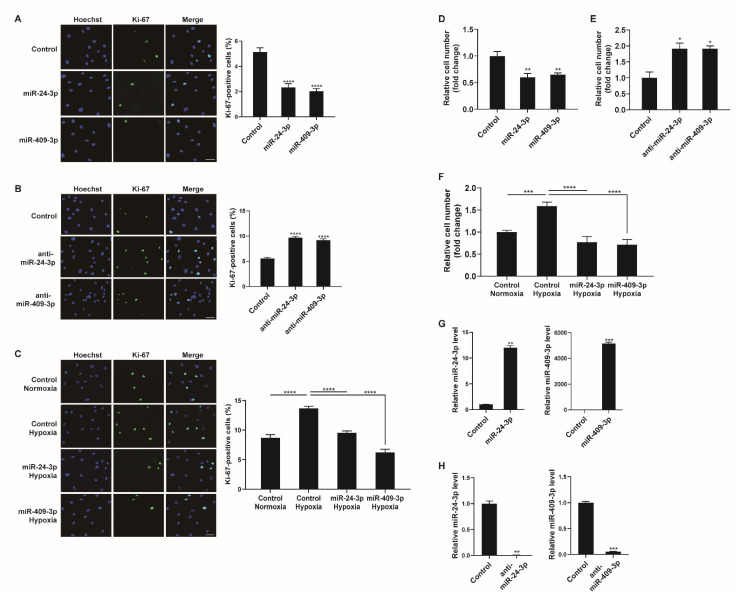
Hypoxia−mediated regulation of miR-24-3p and miR-409-3p affects PASMC proliferation. (**A**,**B**) Representative images of Ki-67 immunostaining of PASMCs transfected with control, miR-24-3p mimic, miR-409-3p mimic, anti-miR-24-3p, or anti-miR-409-3p for 24 h, and calculation of the Ki-67 index. The results are the mean ± S.E. for three independent assays. Scale bar represents 50 μm. (**C**) Immunostaining analysis of PASMCs transfected with control, miR-24-3p mimic, or miR-409-3p mimic after exposure to hypoxia for 24 h. Approximately 200 cells from at least 10 independent fields were counted for each condition, and Ki−67-positive cells are presented as a percentage of the total population. The results are the mean ± S.E. for three independent assays. Scale bar represents 50 μm. (**D**,**E**) PASMCs were transfected with control, miR-24-3p mimic, miR-409-3p mimic, anti-miR-24-3p, or anti-miR-409-3p. Cells were trypsinized and counted using a hemacytometer. The relative number of cells was shown as a fold change. (**F**) PASMCs were transfected with miR-24-3p, or miR-409-3p mimics, followed by hypoxia exposure and the cell counting assay. (**G**,**H**) Levels of the miR-24-3p or miR-409-3p relative to U6 snRNA measured by qRT−PCR in PASMCs transfected with the control, miR-24-3p mimic, miR-409-3p mimic, anti-miR-24-3p, or anti-miR-409-3p for 24 h. The data represent the mean ± S.E. of triplicate measurements of three independent experiments. Statistical analyses were performed using one-way ANOVA Dunnett’s multiple comparisons test (**A**,**B**,**D**,**E**), one−way ANOVA Tukey’s multiple comparisons test (**C**,**F**), and two−tailed unpaired Student’s *t*-test (**G**,**H**). *, *p* < 0.05; **, *p* < 0.005; ***, *p* < 0.0005; ****, *p* < 0.0001.

**Table 1 cells-12-00817-t001:** miRNAs bound to NCL in PASMCs. RNAs from PASMCs were immunoprecipitated with an NCL antibody or IgG, followed by NGS-based small RNA sequencing (GSE184972).

Name	Fold Change (NCL Ab/IgG Pull-down)	*p*-Value
hsa-miR-423-3p	286.670	0.007
hsa-miR-744-5p	186.279	0.002
hsa-miR-24-3p	158.193	0.044
hsa-miR-409-3p	115.211	0.032
hsa-miR-370-3p	111.773	0.001
hsa-miR-92a-3p	97.599	0.001
hsa-miR-1307-3p	54.974	0.036
hsa-miR-193a-5p	53.673	0.000
hsa-miR-382-5p	42.694	0.005
hsa-miR-493-3p	29.618	0.001
hsa-miR-199b-3p	27.346	0.032
hsa-miR-485-3p	21.564	0.013
hsa-miR-323a-5p	18.363	0.023
hsa-let-7d-3p	17.705	0.003
hsa-miR-615-3p	17.234	0.003
hsa-miR-379-5p	14.492	0.005
hsa-miR-335-3p	12.080	0.046
hsa-miR-3074-5p	11.838	0.033
hsa-miR-2682-5p	10.533	0.037
hsa-miR-941	6.806	0.000
hsa-let-7e-5p	6.249	0.035
hsa-miR-4448	5.504	0.002
hsa-miR-132-3p	5.504	0.002
hsa-miR-99b-3p	5.204	0.000
hsa-miR-4745-5p	4.904	0.009
hsa-miR-6892-5p	4.904	0.009
hsa-miR-106b-3p	4.602	0.033
hsa-miR-1972	4.505	0.031
hsa-miR-210-3p	4.205	0.011
hsa-miR-1185-1-3p	4.205	0.011
hsa-miR-1185-2-3p	3.903	0.000
hsa-miR-1180-3p	3.903	0.000
hsa-miR-149-5p	3.600	0.009
hsa-miR-218-5p	2.606	0.010
hsa-miR-30e-5p	2.298	0.010
hsa-miR-342-5p	2.298	0.010
hsa-miR-425-5p	2.298	0.010
hsa-miR-339-3p	2.298	0.010
hsa-miR-6776-3p	2.298	0.010

## Data Availability

The RNA sequencing dataset generated during the current study can be found here: [https://0-www-ncbi-nlm-nih-gov.brum.beds.ac.uk/geo/query/acc.cgi]. The accession numbers for the data reported in this paper are GEO: GSE184972.

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
