# Peer review of "Nucleolin Regulates Pulmonary Artery Smooth Muscle Cell Proliferation under Hypoxia by Modulating miRNA Expression"

_cells, 2023, doi:10.3390/cells12050817_

Round 1

Reviewer 1 Report (New Reviewer)

The authors showed that NCL is downregulated by histone deacetylation under hypoxic conditions in PASMCs. Silencing NCL promotes PASMC proliferation. They also found that this effect of NCL is mediated by interactions with a subset of miRNAs using immunoprecipitation and NGS-based small RNA sequencing.  The identification of NCL-miRNA interactions in hypoxia-induced PASMC proliferation provides a basis for further studies of the molecular mechanisms underlying vascular diseases. The whole manuscript has been well written and scientifically their findings are important. The authors are required to address the following concerns:

1. In Figures 2, 5 and 6, Ki67 is used as a cell proliferation marker. This is not sufficient, and the authors are required to do cell counting and BrdU assays for further confirmation.

2. The figure titles are very confusing and misleading. This includes Figures 1, 3, 4, 5. NCL only interacts some specific miRNAs, but the current titles indicate NCL interacts all the miRNAs. Figure 1, the authors need highlight that NCL is downregulated only in PASMCs.

3. The authors are required to explain why NCL is downregulated only in PASMCs, not in other cell types.

4. In Figure 6,  the authors are required to show NCL expression levels in PASMCs transfected with control, miR-24-3p mimic, miR409-3p mimic, anti-miR-24-3p, or anti-miR-409-3p for 24 h.

5. The authors are required to discuss how to design animal experiments to examine how modulation of NCL or its interacting miRNAs can benefit to the patients with PH using animal models.

Author Response

Reviewer 1

The authors showed that NCL is downregulated by histone deacetylation under hypoxic conditions in PASMCs. Silencing NCL promotes PASMC proliferation. They also found that this effect of NCL is mediated by interactions with a subset of miRNAs using immunoprecipitation and NGS-based small RNA sequencing.  The identification of NCL-miRNA interactions in hypoxia-induced PASMC proliferation provides a basis for further studies of the molecular mechanisms underlying vascular diseases. The whole manuscript has been well written and scientifically their findings are important. The authors are required to address the following concerns:

  1. In Figures 2, 5 and 6, Ki67 is used as a cell proliferation marker. This is not sufficient, and the authors are required to do cell counting and BrdU assays for further confirmation.
    Response: As reviewers 1 and 2 both suggested cell counting, we performed cell counting experiments and added the results to Figures 2 and 6 in the revised manuscript. The results of cell counting experiments corroborated the observation of Ki-67 immunofluorescence staining.
  2. The figure titles are very confusing and misleading. This includes Figures 1, 3, 4, 5. NCL only interacts some specific miRNAs, but the current titles indicate NCL interacts all the miRNAs. Figure 1, the authors need highlight that NCL is downregulated only in PASMCs.
    Response: The titles of Figures 1, 3, 4, 5 and 6 have been changed as suggested by the reviewer.
  3. The authors are required to explain why NCL is downregulated only in PASMCs, not in other cell types.
    Response: As suggested by the reviewer, we added a possible reason why NCL is downregulated only in PASMCs in the Results section.

“According to previous studies, HDAC1 expression levels were elevated in the lungs of patients with idiopathic pulmonary arterial hypertension and rats exposed to hypoxia, and HDAC inhibitors prevented hypoxia-induced pulmonary hypertension [PMID: 22711276; PMID: 32733053; PMID: 17322895]. Therefore, hypoxia is likely to downregulate NCL expression specifically in PASMCs by histone deacetylation.”

  1. In Figure 6, the authors are required to show NCL expression levels in PASMCs transfected with control, miR-24-3p mimic, miR409-3p mimic, anti-miR-24-3p, or anti-miR-409-3p for 24 h.
    Response: As suggested by the reviewer, we checked the expression level of NCL in PASMCs transfected with control, miR-24-3p mimic, miR-409-3p mimic, anti-miR-24-3p, or anti-miR-409-3p for 24 h. As a result, the expression level of NCL was not affected by modulating the amount of these miRNAs. It further supports that hypoxia-induced changes in NCL levels affect PASMC proliferation by regulating expression levels of specific miRNA.
  2. The authors are required to discuss how to design animal experiments to examine how modulation of NCL or its interacting miRNAs can benefit to the patients with PH using animal models.
    Response: As suggested by the reviewer, we mentioned further studies using animal models in the Discussion section.

“To explore the potential therapeutic benefits of NCL or interacting miRNAs on pulmonary hypertension, it is necessary to investigate whether modulation of NCL or interacting miRNAs is effective in attenuating pulmonary vascular remodeling in animal models, such as a chronic hypoxia-induced rat model.”

Reviewer 2 Report (New Reviewer)

The present study plausibly demonstrated the causative role of the down-regulation of nucleolin (NCL) in the hypoxia-induced increase in proliferative activity of PASMC and convincingly elucidated some underlying mechanisms that involve the regulation of miRNA abundance presumably mediated by interaction of DGCR8 or Ago2  with NCL. How miRNAs that are regulated by NCL contribute to the proliferative activity of PASMC remains to be determined; however, this problem could be a subject of future research. The experiments are well-designed and performed, the data are sound and clear, the conclusion is well supported by the data and the manuscript is well prepared in grammatically sound language. The limitation of the study is a lack of physiological relevance of the findings in in vivo situation. That the detection of Ki67 is the only index of proliferation is another drawback of the present study.

Specific points

1. The detection of Ki67 is the only index of cell proliferative activity that was employed in the present study. The possibility that NCL-miRNA axis just regulate the expression of Ki67, but not cell proliferation, is recommended to be ruled out. Some other more functional assay of proliferation, such as direct cell counting or MTT assay, is recommended to performed to corroborate the observation of Ki67 immunofluorescence staining. The roles of hypoxia and NCL are recommended to be confirmed with these assays.

2. The reproducibility is one of the most important matters in science. The figure legends state that the date are the means +/- SE of triplicate, while the Statistical analysis section of the Materials and Methods states that all experiments were performed with at least three independent repetitions. "Triplicate" usually indicates three measurements in one set of experiment. Therefore, there is some uncertainty and doubt about the confirmation of reproducibility by multiple sets of experiments. Please clarify this uncertainty and doubt. The supplemental information only show the original image of the representative data. Showing not only representative but all images could be a proof of reproducibility.

3. Two methods of hypoxia are described in the Cell culture and hypoxia section of the Materials and Methods. Namely, one method is to culture cells under 1% O2 condition, while the other is to subject cells to CoCl2 challenge. However, it is unclear which condition was used in each  experiment. Please clarify in each  legend and in the relevance part of the text which condition was used. Moreover, if the condition is arbitrary varied depending on the experiment, the rationale for such selective employment should be explained.

Author Response

Reviewer 2

The present study plausibly demonstrated the causative role of the down-regulation of nucleolin (NCL) in the hypoxia-induced increase in proliferative activity of PASMC and convincingly elucidated some underlying mechanisms that involve the regulation of miRNA abundance presumably mediated by interaction of DGCR8 or Ago2  with NCL. How miRNAs that are regulated by NCL contribute to the proliferative activity of PASMC remains to be determined; however, this problem could be a subject of future research. The experiments are well-designed and performed, the data are sound and clear, the conclusion is well supported by the data and the manuscript is well prepared in grammatically sound language. The limitation of the study is a lack of physiological relevance of the findings in in vivo situation. That the detection of Ki67 is the only index of proliferation is another drawback of the present study.

Specific points

  1. The detection of Ki67 is the only index of cell proliferative activity that was employed in the present study. The possibility that NCL-miRNA axis just regulate the expression of Ki67, but not cell proliferation, is recommended to be ruled out. Some other more functional assay of proliferation, such as direct cell counting or MTT assay, is recommended to performed to corroborate the observation of Ki67 immunofluorescence staining. The roles of hypoxia and NCL are recommended to be confirmed with these assays.
    Response: As suggested by the reviewer, we performed cell counting experiments and added the results to Figures 2 and 6 in the revised manuscript. The results of cell counting experiments corroborated the observation of Ki-67 immunofluorescence staining. In addition, it was further confirmed through cell counting analysis that cell proliferation was increased by hypoxia and decreased by NCL.
  2. The reproducibility is one of the most important matters in science. The figure legends state that the date are the means +/- SE of triplicate, while the Statistical analysis section of the Materials and Methods states that all experiments were performed with at least three independent repetitions. "Triplicate" usually indicates three measurements in one set of experiment. Therefore, there is some uncertainty and doubt about the confirmation of reproducibility by multiple sets of experiments. Please clarify this uncertainty and doubt. The supplemental information only show the original image of the representative data. Showing not only representative but all images could be a proof of reproducibility.
    Response: We clarified the figure legends as follows. “Results are the mean ± S.E. of triplicate measurements of three independent experiments.” In addition, we uploaded supplemental information containing more original images of replicate experiments.
  3. Two methods of hypoxia are described in the Cell culture and hypoxia section of the Materials and Methods. Namely, one method is to culture cells under 1% O2condition, while the other is to subject cells to CoCl2 challenge. However, it is unclear which condition was used in each experiment. Please clarify in each legend and in the relevance part of the text which condition was used. Moreover, if the condition is arbitrary varied depending on the experiment, the rationale for such selective employment should be explained.
    Response: We found an error in the Cell culture and hypoxia section of the Materials and Methods and apologize for any misunderstanding. In this study, hypoxia was induced with only 1% O2 treatment. Since hypoxia induction experiments by CoCl2 treatment were not performed in this study, the incorrectly written sentence in the Materials and Methods was deleted.

Round 2

Reviewer 1 Report (New Reviewer)

The authors have well addressed most of my questions, except question 3.

Author Response

Response to Reviewers

Reviewer 1

The authors have well addressed most of my questions, except question 3.

Response: In our last revision, we mentioned that NCL in PASMCs may be downregulated due to changes in HDAC1 expression. This revision adds an explanation as to why NCL reduction occurs only in PASMC.

“The unique responsiveness of PASMCs to hypoxia has been reported [PMID: 21368105]. Hypoxia increases the proliferation of PASMCs, whereas it inhibits proliferation in many other cells. PASMC-specific reduction of NCL expression may contribute to inducing the unique responsiveness of PASMCs to hypoxia.”

Question 3. The authors are required to explain why NCL is downregulated only in PASMCs, not in other cell types.
Original Response: As suggested by the reviewer, we added a possible reason why NCL is downregulated only in PASMCs in the Results section.

“According to previous studies, HDAC1 expression levels were elevated in the lungs of patients with idiopathic pulmonary arterial hypertension and rats exposed to hypoxia, and HDAC inhibitors prevented hypoxia-induced pulmonary hypertension [PMID: 22711276; PMID: 32733053; PMID: 17322895]. Therefore, hypoxia is likely to downregulate NCL expression specifically in PASMCs by histone deacetylation.”

Reviewer 2 Report (New Reviewer)

The manuscript has been satisfactorily revised. There is no further point to be addressed.

Author Response

Reviewer 2

The manuscript has been satisfactorily revised. There is no further point to be addressed.

Response: I appreciated the constructive criticism of the reviewer once again.

This manuscript is a resubmission of an earlier submission. The following is a list of the peer review reports and author responses from that submission.

Round 1

Reviewer 1 Report

The manuscript by Lee and Kang is presented in a straightforward way and identifies nucleolin as a modulator of PASMC miRNA expression and proliferation.

While the conclusions of the authors are to the most part supported by their data, the overall relevance of this study appears to be limited:

1. All results are solely based on cultured PASMCs from one donor (n=1) and experiments were performed in triplicate. PASMCs from multiple donors (e.g. n=5) should be utilized to underline the robusness of the authors findings.

2. The impact of nucleolin on SMC proliferation (as well as pulmonary artery hypertension) is already known (e.g. PMID: 33219625, 31893573, 32587339).

3. The rationale of this study remains unclear as short-term hypoxia induces PASMC constriction in first place. Chronic hypoxia promotes PASMC proliferation, which, however, is triggered by growth factors (e.g. PDGFB) and cytokines partially released from lung endothelial cells.  

Specific comments:

- Introduction: "Hypoxia stimulates the abnormal proliferation and migration of vascular smooth muscle cells (VSMCs), resulting in the accumulation of proliferating neointimal VSMCs, which is a prominent feature of several vascular diseases, such as hypertension and atherosclerosis [8, 9]." The generalization of the context discussed in the citated literature is misleading. As PASMCs are utilized throughout this study, the authors are encouraged to introduce responses of (PA)SMC in the hypoxic lung more carefully.

- Materials and Methods, statistical analysis:
Please define the type of the ANOVA posthoc and  Student's t-test. How was normality of the data and equality of variances across comparison groups tested?

- Results:
The statistical tests throughout the whole study produce always similar results (p<0.05). Please explain.

Quantification of the results shown in Figure 4C-F is missing.

Reviewer 2 Report

VSMC in different arteries are very different and care should be taken in distinguishing PASMC from VSMC in other beds (Mitochondrion. 2021 Mar;57:97-107. doi: 10.1016/j.mito.2020.11.012)  This study uses only PASNC and provides data very relevant to the pulmonary circulation but it is not clear that these same pathways operated in other vascular smooth muscle cells. This should be clarified throughout the introduction and the discussion and should be designated as PASMC, not VSMC.

In the study designed to see if hypoxia regulation of NCL was unique to PASMC, using microvascular smooth muscle cells, aortic smooth muscle cells or other systemic VSMCs would be more informative than using the transformed cell lines such as HeLa and HEK293 cells.  I suggest the authors look at this pathway in other VSMC to see if it is unique to PASMC or universal throughout all VSMC.

References 53 and 54 (third paragraph on Pg 15) do not discuss “hypoxia-induced lesions” and I am not certain what is meant by this statement.  53 discusses differences in arterial and venous smooth muscle responses to hypoxia while 54 describes proliferation on pulmonary artery smooth muscle cells during chronic hypoxia-9induced hypertension.  These are indeed vascular pathologies but remodeling seems like a better descriptor than lesions.